# Consequences of the Early Phase of the COVID-19 Pandemic for Home-Healthcare Recipients in Norway: A Nursing Perspective

**DOI:** 10.3390/healthcare11030346

**Published:** 2023-01-25

**Authors:** Line Melby, Merete Lyngstad, Solveig Osborg Ose

**Affiliations:** 1Department of Health Research, SINTEF Digital, 7465 Trondheim, Norway; 2Norwegian Nurses Association, 0104 Oslo, Norway

**Keywords:** COVID-19, nursing, home healthcare, isolation, health concerns, preparedness plans, crisis management

## Abstract

Municipal home-healthcare services are becoming increasingly important as growing numbers of people are receiving healthcare services in their home. The COVID-19 pandemic represented a challenge for this group, both in terms of care providers being restricted in performing their duties and care receivers declining services for fear of being infected. Furthermore, preparedness plans were not always in place. The purpose of this study is to investigate the consequences for recipients of home healthcare in Norway of the actual level of COVID-19 infection spread in the local population, as observed by licensed nurses working in home-healthcare services. Approximately 2100 nurses answered the survey. The most common adverse consequences for home-healthcare recipients were increased isolation and loneliness, increased health concerns, and the loss of respite care services. An increased burden for relatives/next of kin and fewer physical meetings with service providers were frequently observed and reported as well. This study shows that there were more adverse consequences for service users in municipalities with higher levels of contagion than in those with lower levels of contagion. This indicates that the municipalities adapted measures to the local rate of contagion, in line with local municipal preparedness strategies.

## 1. Introduction

When the COVID-19 pandemic struck in the winter of 2020, the world was facing a situation of great uncertainty. There was little knowledge about how the pandemic would develop and what the consequences would be for both individuals and society at large. What was known was that most countries had a limited capacity to handle a situation with many patients with extensive healthcare needs. To reduce the spread of COVID-19 and to prevent a collapse of healthcare services, many countries around the world implemented lockdowns and took strict infection control measures. Such measures impacted all parts of healthcare services, affecting both staff and patients. Much attention was paid to patients in hospitals; particular attention was paid to the follow up of COVID-19 patients, but also to non-COVID patients who experienced restrictions on visitations and strict infection control measures. Persons living at home and receiving home-healthcare services were also affected, but appeared to be less visible in the public debate. In this paper, we draw attention to the consequences of the first phase of the pandemic for home-care recipients, defined as home-dwelling persons receiving home-healthcare services.

Due to a growing ageing population with increasing needs for healthcare services, a lack of capacity in hospitals, and incentives for the early discharge of patients from hospitals, the use of home-healthcare services has increased over the past decade [1]. This trend is likely to further increase in the years to come, and it is therefore important to strengthen home-healthcare services and ensure that the care recipients experience is high-quality and safe [2]. Healthcare in Norway is mainly publicly financed, and all citizens have the right to healthcare services, which includes home healthcare. The municipalities are responsible for home healthcare, which is staffed by nurses, auxiliary nurses, and assistants. The services provided by home care staff are varied, ranging from preparing meals and coordinating healthcare services across actors to performing advanced medical procedures, such as administering intravenous treatments. Following the trend in which an increasing number of services have been transferred from hospitals to home healthcare, increasingly advanced healthcare is now provided in the individual's own home. Home-care recipients may have visits from home healthcare staff as often as five times a day, while some are seen only once a week for medication handling.

To summarise, home healthcare is becoming an increasingly important service, attending to medical, practical, and organisational tasks [3]. In times of disaster, these services are of vital importance for vulnerable, home-dwelling persons [4], and efforts should therefore be made to sustain the services throughout such times [5].

### 1.1. National and Local Crisis Management 

Handling a pandemic is a prime example of crisis management, and emergencies such as the pandemic must be handled on both the national level and at the local, municipal level [6]. Within policy research, the evaluation of whether a crisis is handled in a ”successful” way [7] and what constitutes “success” is addressed [8,9]. As a framing for our study, an understanding of the elements and challenges of crisis management (and the study of it) is useful. McConnell and Stark [10] outline the following six key issues, which are particularly related to policy responses to COVID-19:Problem framing: narrowing down a social phenomenon to a “problem” and making assumptions about its severity;The relations between government and experts: navigating when knowledge about the virus is uncertain and when experts’ advice is sometimes conflicting. How much trust can be placed in experts?The public sector’s preparedness for crisis: lacking or insufficient crisis management plans and the need for the adaptation to and the improvisation of pandemic plans;One-size-does not-fit-all solutions: variations in policies and responses across nations and regions; variations in the flow of power during crises, sometimes favouring centralised approaches, other times favouring decentralised ones;Pluralism in response to crises: societies are always pluralistic, and this does not disappear during crisis; consensus vs. pluralism in opinions on the crisis and its management;Social inequality and vulnerability during a crisis: both the crisis itself and the response measures affect people differently.

There is no definitive benchmark for crisis management success [8], and there can be many standards against which crises measures are judged. Inevitably, crises management measures will produce winners and losers. For example, the lockdown during the pandemic has reduced the numbers of infected persons; however, it has also negatively affected businesses and cultural life [11]. Healthcare services and healthcare recipients may have been affected in different ways.

### 1.2. The Case of Norway

Crisis management is built on predefined emergency plans and, during the pandemic, a national (central) emergency plan and local plans in the municipalities co-existed in Norway. However, from the early onset of the pandemic, it was clear that these plans were not specifically developed for handling a situation such as the COVID-19 pandemic. Particularly relevant for understanding how home-healthcare services were affected is the fact that the Infection Control Act places infectious disease expertise at the local government level [12]. This implies that the municipalities, as local governments represented by an elected municipal council (or the chief medical officer in the municipality in case of a health crisis), can implement a range of local measures, including local quarantines, travel restrictions, assembly restrictions and the closure of kindergartens, schools, businesses, and events [ibid]. The decision as to whether it is the central or local government's responsibility to regulate actions and ensure adherence to laws and plans has varied during the course of the pandemic. In the first phase of the pandemic in Norway (March 2020–October 2020), which is addressed in this article, the municipalities played a major role in crisis management. As the municipalities developed their own laws and rules, the crisis was handled differently in the various municipalities [11,13].

When the pandemic broke out, the Norwegian central government implemented a lockdown in March 2020. Several measures were taken to flatten the infection curve and keep the healthcare services at a sustainable level. Central government recommendations were issued regarding hand hygiene and cough etiquette, working from home, if possible, that public transport should be avoided, that domestic travel should be limited, and that there should be no visits to members of vulnerable groups in health institutions [14]. To secure bed capacity in the hospitals, the threshold for admissions became higher. The hospitals were closed to visitors. The specialist health service also discharged many patients to the municipal healthcare service earlier than normal [11]. This entailed a double burden for the healthcare services in the municipalities, which had to both change and scale up their own services as well as take care of sicker persons with even more complex needs than before. In parallel with the national measures, the municipalities started to develop and implement their own local measures.

International studies indicate that persons receiving home-healthcare services experienced negative consequences from the pandemic crisis measures. Home-healthcare recipients had their services postponed, reduced, or removed to both prevent the spread of infection and to ensure preparedness [15,16,17,18]. Furthermore, in Norway, there was a lack of personal protective equipment (PPE) in the home-healthcare services, and visits from home-healthcare staff were therefore restricted to a minimum. Many persons isolated themselves and cancelled visits from home healthcare. Many were also isolated from their families and next of kin. It has been reported that this led to dysfunction and possible loss of health [19,20]. The changes and reductions of services that the pandemic led to for home-healthcare recipients should also be seen in light of an increasing prevalence of missed care in municipal healthcare services in general. Several studies have pointed out that care is left undone due to low staffing levels, unfavourable working environments, and a lack of resources [21,22,23,24]. The pandemic crisis measures may have intensified the frequency of missed care.

In addition to consequences for care recipients, healthcare workers were also negatively affected by the pandemic. As a consequence of working with limited resources and knowledge of the virus, nurses experienced emotional exhaustion, depressive symptoms, and reduced personal accomplishments [25,26].

In Norway, it has been discussed in hindsight whether the authorities overreacted to the situation in the first phase of the pandemic through stages of building up negative emotions and the propagation of fear and isomorphic decision-making, leading to an intractable crisis [27]. If central and local authorities overreacted to the situation, the reported negative consequences of the restrictions might be independent of the actual level of contagion.

The purpose of this study is thus to investigate how the COVID-19 infection level in Norwegian municipalities correlated with the extent of the reported consequences for home-healthcare recipients.

## 2. Materials and Methods

To study whether the severeness of the pandemic (measured by confirmed COVID-19 cases relative to population size) was in line with the degree of reported negative consequences for home-healthcare recipients (reported by the licensed nurses working in the services), we tested whether high levels of contagion were found in municipalities with more reported consequences; that is, if the reported negative consequences vary according to the actual local levels of contagion.

We used a cross-sectional approach with a questionnaire administered between September 2020 and October 2020. Licensed nurses reported on 23 adverse consequences for home-healthcare services users in whichever of the 356 municipalities they were employed in. The date of survey completion was automatically registered, and the date of completion and municipality information were used to link the number of accumulated, confirmed COVID-19 cases.

### 2.1. Setting 

This study was set across locations of local home-healthcare services in Norway, and the target provider group was licenced nurses. In 2020, when the data for this study were collected, 160,673 persons received home-healthcare services, according to Statistics Norway. Staff in home healthcare worked in an ambulatory setting in which they visited persons in need of healthcare services in their homes.

### 2.2. The Questionnaire

We searched the literature but did not find any questionnaires that measured nurses’ experiences of the consequences of the COVID-19 pandemic for patients and their next of kin. Therefore, we developed a questionnaire in several phases. First, the researchers outlined the main themes that we wanted to investigate. Second, we organised an expert group with nurses from the professional groups of the Norwegian Nurses Organisation (NNO). These nurses had clinical experience, and some of them had research skills. Some of the nurses provided written feedback relating to the themes that had been sent to them via e-mail. Thereafter, two of the researchers met with the expert group in a Teams meeting to further discuss the questions. Third, we organised a group of persons which represented patient and next of kin associations to gain input on the themes and questions which they considered important. The group consisted of persons from five associations representing persons that are frequently among recipients of home healthcare service. The following groups were represented: persons with chronic diseases, persons with mental illnesses, and persons with disabilities. At this point in time, the associations had received a good deal of input from their members about their experiences with the pandemic, and this information was valuable for the design of the questionnaire. The meeting with the patient and next-of-kin association was held digitally, via Teams. Two of the researchers and persons from the NNO took part.

The next phase consisted of the questionnaire development. The questionnaire was developed based on input from the aforementioned meetings. It was thereafter pilot-tested by the expert group and licensed nurses who worked in healthcare services. It was then revised. These steps provided significant and valuable information for what became the final version. The final version of the questionnaire contained 23 items and included information about the municipalities in which the nurses worked.

The questionnaire items were rated on a five-point Likert scale as follows: (1) strongly disagree, (2) disagree, (3) neither agree nor disagree, (4) agree, and (5) strongly agree. In addition, free-form answers were used to gain more information from the respondents. A total of 58 respondents commented in free text. The Questionnaire containing 23 items can be found in the Appendix A. 

### 2.3. Sample and Participation

All nurses who were members of the Norwegian Nurses Organisation and had registered an e-mail address in the member database were invited to participate. The inclusion criterion was that they should have been employed in the period that commenced with beginning of the pandemic (March 2020). An exclusion criterion was not being employed in the healthcare sector. Among those who were employed in the healthcare sector, 35% responded (*N* = 26,915). From this sample, we established a sub-sample of 2860 licenced nurses working with adults or elderly service users in home care services. Between 2072 and 2116 nurses (72–74%) responded to the 23 questions regarding consequences for service users.

### 2.4. Data Collection and Linkage

The data collection period covered 28 days (22 September to 19 October 2020). In addition to the 23 questions about consequences for service users, we collected information about the municipalities in which the workplaces of the responding nurses were located. Based on this information, we linked information about the reported number of accumulated COVID-19 cases in each municipality from the Norwegian Surveillance System for Communicable Diseases (MSIS) on each of the 28 days. Thus, we had information about the consequences for the adult service users in the view of the nurses working in home care services as well as the actual dissemination of disease in the municipality in which the workplace of each nurse was located. The MSIS data were provided by the Norwegian Institute on Public Health, and the municipality data were extracted from Github on each date and subsequently linked to the answers from the survey using municipality numbers, which entailed a four-digit code assigned to each of the 356 municipalities.

### 2.5. Data Analyses

We treated each of the 23 consequences for the service users as different dependent variables. The dependent variables were ordinal in nature; that is, they had a natural ordering with values from “To a very large extent” to “To a very small extent”, so we estimated ordinal regressions for which the only explanatory variable was the accumulated number of confirmed COVID-19 cases per 1000 inhabitants on the response date in the municipality where the workplace was located. The ordered probit model provided an appropriate fit to these data, preserving the ordering of response options while making no assumptions on the interval distances between the response options. To test whether the level of contagion in the municipality where the responding nurse worked was correlated with the extent of the reported consequences for the service users, we used a standard chi-squared test for each of the 23 consequences separately, in which the null hypothesis was that the regression coefficients in the model were equal to zero. Pairwise correlation coefficients between the consequences (C1–C23) are shown in the Appendix A. 

Graphical representations of the results of the linear predictions of each of the 23 consequences are displayed with 95% confidence intervals around the mean.

### 2.6. Ethical Considerations

The nurses who were invited to answer the questionnaire received written information explaining that their participation was anonymous and that returning the questionnaire meant that they agreed to participate in the study. For the questionnaire, a secure sockets layer (SSL) was used to provide security to the data when transferred between the web browser and the server. All data were securely stored and protected using approved solutions. This was a prerequisite for study approval. The methods for the data collection and the handling of interviews, as well as the questionnaire, were approved by the Norwegian Centre for Research Data (registration number 580244) and the Research Ombudsman in SINTEF.

## 3. Results

The most frequently reported negative consequence for persons receiving home care services, as reported by the nurses, was increased isolation and loneliness. A total of 79.2% reported that this was experienced to a very large or large extent; see Table 1. The second most frequent consequence was an increase in health concerns (60.5%), while loss of respite care services was the third most frequent consequence reported (47.2). an increased burden on relatives or the next-of-kin and fewer physical meetings with the services were also reported by many of the respondents, 35.7% and 35.5%, respectively (to a very large or to a large extent).

Answers to the open-ended questions support these findings. One nurse wrote: ‘It has been–and is–very difficult for many patients and next of kins, that day care and respite care were closed or reduced. Several patients have had a reduced quality of life, become physically poorer, and have struggled with anxiety and depression. More use of drugs in vulnerable patients. Several patients had to be admitted to hospitals because they could not eat and drink enough, when they did not sit with others around the meals at the day care centre. During the summer, their state of health was so reduced that they were admitted. Also observed that there was more focus on pain and discomfort in some, and thus higher consumption of medication than when they were more active’.

Few respondents reported that the service users to a very large/large extent stopped taking medication because they the medications were immunosuppressive (0.7%). However, as can be seen in the fifth column of Table 1, 38.3% answered that this consequence was “not relevant”, which suggests that such medications were not used by many of the home-care-service users. Additionally, few (2.8%) (to a very large or large extent) reported that acute situations occurred. 

As can be seen from the correlation matrix provided as Appendix A, the strongest correlations were found between a deterioration of condition and poorer prognosis (correlation = 0.73) and between delayed diagnostics and delayed follow-up/treatment (correlation = 0.72).

As can be seen from the chi-squared test in the last column of Table 1, most of the extent of consequences were related to the actual level of contagion in the municipality where the nurses worked. This result implies that most of the consequences to the home-care-service users were in line with the actual level of contagion in the geographical area of residence for the service users. As can be seen in Table 2, all coefficients are negative, indicating that consequences were observed to a smaller extent (increased value of consequences) in less-infected municipalities (negative coefficients). To put this another way, nurses that were working in more infected areas or municipalities reported more consequences for the users of home care services.

Consequences that did not seem to depend on the level of contagion in the first part of the pandemic included an increased contact with services, ceasing to take medications because they were immunosuppressive, and problems with access to medication. The results imply that these consequences happened independently from the actual level of contagion and that these consequences were perhaps more national in nature than the other consequences included.

As is shown in Table 2 and confirmed by the visualisation of the results in Figure 1, the consequences most dependent on the level of the local level of contagion were the deterioration of condition (b = −0.073), adverse events and reduced level of functioning (b = −0.07), and poorer prognosis (b = −0.064).

An element of uncertainty in the response categories of the survey is the response ‘not relevant’. This could mean that the consequence was not observed at all or that none of the home care service users could have experienced this consequence. In the data and analyses presented in Table 1 and Table 2 and Figure 1, we have assigned code 6 to this answer, indicating that this is less than “To a very small extent.” The alternative is to treat this answer as a missing value and explore whether the results are changed. This process has been performed, and we found that the results are robust and are not altered qualitatively.

## 4. Discussion

In this study, we have investigated the consequences of the COVID-19 pandemic situation on home-healthcare recipients in Norway. We were interested in finding out if the municipalities’ actual level of COVID-19 infection was reflected in the frequency of adverse consequences for home-healthcare recipients. 

Regardless of the infection level in the municipality, the most common adverse consequences observed were increased isolation and loneliness as well as increased health concerns. This is in line with previous studies that found such experiences to be consequences of both the elimination of formal services and a reduction in informal caregiving [19,20]. 

Additionally, among the most frequently observed consequences in our study was a loss of respite care services. Furthermore, increased burdens on relatives/next-of-kin and fewer physical meetings with home services were frequently occurring consequences. A reduction in and the termination of home care and respite services were direct consequences of the municipalities’ efforts to reduce the spread of the SARS-CoV-2 virus. In turn, this had second-order consequences for both home-care recipients and informal carers. In their study, Vislapuu et al. [18] found similar consequences from restrictions imposed by pandemic measures. They investigated formal and informal care utilisation among home-dwelling persons with dementia and found that both types of care were reduced, resulting in a dramatic change in the care situation for this group. Other studies [16,20] also pointed to a reduction in formal services. This reduction, in turn, imposed an additional burden on informal caregivers in, for example, France [28] and the United Kingdom [29].

By linking information about observed consequences and the actual level of contagion in the municipality where each nurse worked, we find that more adverse consequences for care recipients were observed in municipalities with higher levels of contagion. Only for three of the mapped consequences do we not find a systematic correlation with the actual level of contagion; these include needing more contact with the services than before the corona situation, ceasing to take medications because the medications are immunosuppressive, and having problems with access to medication. These findings indicate that scepticism towards taking immunosuppressive medication and access to medication are consequences stemming from national-level policies rather than consequences due to local restrictions and levels of contagion.

Home-healthcare services were important services during the pandemic, both for the early recognition of COVID-19 cases and for the continuity of care and prevention [4,30]. The majority of home-healthcare recipients in Norway are elderly persons, and they were particularly affected when the services were reduced in the municipalities. Previous studies have shown that older adults in general have been found to be disproportionately impacted by the COVID-19 pandemic [31,32]. Our study lends support to the fact that elderly persons who receive home healthcare were severely affected by the pandemic, and points to the specific consequences they faced. The reduction or temporary stop of services intensify an already challenging situation in the municipalities, where missed care is frequently occurring [21,33].

Our study was not designed to show in detail how the various municipalities responded to the crisis that the pandemic led to. However, we have shown that municipalities seem to have adapted their measures to the level of contagion they faced. Relating to the crisis-management literature, it is clear that the municipalities, as other governmental bodies, have been forced to compromise and make trade-offs in their approach to the COVID-19 pandemic. The “problem framing” of the pandemic [10]—that is, the effects of the virus–was communicated early on by the media and the central health authorities. Experts and expert knowledge have a central place during crises and are used by policy makers in the formulation of evidence-based policies [6]. In Norway, the expert bodies' advice was often quite cautious, but politicians decided to take stronger measures because they balanced a wider range of considerations [10,11,13]. In combination with a lack of PPE, this may have led to stricter measures, specifically a reduction in and the elimination of home care services and respite care, than might have been necessary.

An important factor in handling a crisis such as a pandemic at the municipal level is the existence of local pandemic-preparedness plans [9,10]. Such plans did not exist. This led to improvisation and continuous adaptations to the pandemic’s development and to nationally formulated regulations. Such an approach represents typical policy responses during a crisis, in which one can observe a shift between local and central approaches [10]. For the municipalities, supported by findings in our study, this meant that they adapted their pandemic response to the local level of contagion.

Responding to the COVID-19 pandemic was not easy for authorities on any level; nor was it easy for the home-care service managers, given that there was a lack of evidence-based knowledge and much uncertainty regarding the efficacy of measures used to fight the pandemic. Overall, the main decision-making style and handling of the pandemic outbreak might be said to be based on a pragmatic collaborative approach combining argumentation and feedback, flexibility, and compromise [7]. Such an approach makes sense given the fundamental uncertainty of the situation. It would have been impossible to avoid negative consequences for any group; with respect to home care services, the recipients and their informal carers certainly experienced their own.

## 5. Conclusions

Adverse consequences for adult home-healthcare service users in the first phase of the COVID-19 pandemic were observed by licenced nurses working in home-healthcare services. Most of the observed consequences for service recipients were correlated with the actual level of contagion in the municipality where the responding nurse worked. This indicates that the local authorities took measures to try to slow the spread of the virus and protect the health and safety of their citizens according to the local infection situation; therefore, more adverse consequences are observed in municipalities with high levels of contagion than in municipalities with lower levels of contagion.

## Figures and Tables

**Figure 1 healthcare-11-00346-f001:**
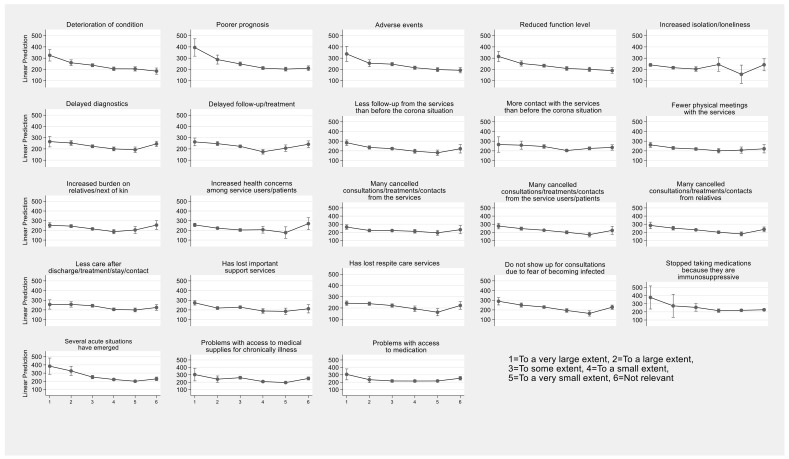
Predictions of adverse consequences for home-healthcare service recipients with 95% CIs around the mean level of number of reported COVID-19 cases pr. 100,000 inhabitants.

**Table 1 healthcare-11-00346-t001:** Consequences for home-healthcare recipients of pandemic restrictions and crisis measures, as reported by nurses in the services. Likelihood ratio chi-squared test of the observed consequences and the accumulated number of positive COVID-19 tests per 1000 inhabitants in the municipality of the home healthcare service.

	To a Very Large/ Large Extent n (%)	To Some Extent n (%)	To a Small/ Very Small Extent n (%)	Not Relevant n (%)	Total	Likelihood Ratio Chi-Squared Test c^2^ (df) *p*-Value
Deterioration of condition	259 (12.3)	753 (35.6)	918 (43.4)	183 (8.7)	2113	37.7 (1) *p* < 0.001
Poorer prognosis	110 (5.3)	476 (22.7)	1203 (57.4)	306 (14.6)	2095	29.0 (1) *p* < 0.001
Adverse events	176 (8.4)	620 (29.5)	1056 (50.3)	247 (11.8)	2099	34.8 (1) *p* < 0.001
Reduced level of functioning	328 (15.6)	739 (35.2)	839 (39.9)	196 (9.3)	2102	34.5 (1) *p* < 0.001
Increased isolation/loneliness	1675 (79.2)	333 (15.7)	59 (2.8)	49 (2.3)	2116	7.6 (1) *p* = 0.006
Delayed diagnostics	353 (16.8)	785 (37.3)	702 (33.4)	263 (12.5)	2103	5.0 (1) *p* = 0.025
Delayed follow-up/treatment	544 (25.9)	925 (44)	478 (22.7)	156 (7.4)	2103	12.3 (1) *p* < 0.001
Less follow-up from the services than before the corona situation	634 (30)	906 (42.9)	500 (23.7)	72 (3.4)	2112	26.1 (1) *p* < 0.001
More contact with the services than before the corona situation	103 (4.9)	404 (19.2)	1412 (67.2)	182 (8.7)	2101	1.4 (1) *p* = 0.235
Fewer physical meetings with the services	749 (35.5)	867 (41.1)	419 (19.8)	76 (3.6)	2111	10.6 (1) *p* = 0.001
Increased burden on relatives/next-of-kin	753 (35.7)	898 (42.5)	393 (18.6)	68 (3.2)	2112	12.9 (1) *p* < 0.001
Increased health concerns among service users/patients	1278 (60.5)	645 (30.5)	152 (7.2)	37 (1.8)	2112	14.4 (1) *p* < 0.001
Many cancelled consultations/treatments/contacts from the services	668 (32)	942 (45.2)	420 (20.1)	56 (2.7)	2086	6.9 (1) *p* = 0.009
Many cancelled consultations/treatments/contacts from the users/patients	472 (22.6)	1006 (48.2)	551 (26.4)	57 (2.7)	2086	26.0 (1) *p* < 0.001
Many cancelled consultations/treatments/contacts from relatives	343 (16.5)	822 (39.5)	757 (36.4)	157 (7.6)	2079	18.9 (1) *p* < 0.001
Less care after discharge/treatment/stay/contact	231 (11.1)	626 (30.2)	1050 (50.7)	165 (8)	2072	14.3 (1) *p* < 0.001
Has lost important support services	897 (43.2)	732 (35.2)	373 (18)	75 (3.6)	2077	19.2 (1) *p* < 0.001
Has lost respite care services	983 (47.2)	623 (29.9)	362 (17.4)	116 (5.6)	2084	15.7 (1) *p* < 0.001
Does not show up for consultations due to fear of becoming infected	396 (19)	836 (40.1)	562 (27)	289 (13.9)	2083	20.5 (1) *p* < 0.001
Stopped taking medications because they are immunosuppressive	14 (0.7)	64 (3.1)	1203 (57.9)	796 (38.3)	2077	0.1 (1) *p* = 0.787
Several acute situations have emerged	59 (2.8)	312 (15)	1421 (68.4)	284 (13.7)	2076	15.8 (1) *p* < 0.001
Problems with access to medical supplies for chronic illness	95 (4.6)	373 (18)	1281 (61.7)	328 (15.8)	2077	3.3 (1) *p* = 0.068
Problems with access to medication	110 (5.3)	501 (24.1)	1230 (59.2)	238 (11.4)	2079	0.5 (1) *p* = 0.466

**Table 2 healthcare-11-00346-t002:** Results from ordered probit regressions. Consequences: to a very large extent (1) … to very small extent (5) and not relevant (6) explained by reported COVID-19 cases per 1000 inhabitants.

	Coef.	Std. Err.	*z*	*p* > *z*	[95% Conf.]	[Interval]
Deterioration of condition	−0.073	0.0119	−6.14	0	−0.0963	−0.0497
Poorer prognosis	−0.064	0.0120	−5.38	0	−0.0879	−0.0410
Adverse events	−0.070	0.0120	−5.90	0	−0.0939	−0.0471
Reduced level of functioning	−0.070	0.0119	−5.87	0	−0.0932	−0.0465
Increased isolation/loneliness	−0.035	0.0126	−2.76	0.006	−0.0592	−0.0100
Delayed diagnostics	−0.027	0.0119	−2.25	0.025	−0.0501	−0.0034
Delayed follow-up/treatment	−0.042	0.0120	−3.50	0	−0.0654	−0.0185
Less follow-up from the services than before the corona situation	−0.061	0.0119	−5.10	0	−0.0843	−0.0375
Need for more contact with the services than before the corona situation	−0.014	0.0119	−1.19	0.235	−0.0376	0.0092
Fewer physical meetings with the services	−0.039	0.0119	−3.25	0.001	−0.0619	−0.0153
Increased burden on relatives	−0.043	0.0119	−3.59	0	−0.0662	−0.0195
Increased health concerns among users/patients	−0.046	0.0121	−3.79	0	−0.0698	−0.0222
Many cancelled consultations/treatments/contacts from the services	−0.032	0.0120	−2.62	0.009	−0.0552	−0.0080
Many cancelled consultations/treatments/contacts from the users/patients	−0.062	0.0121	−5.10	0	−0.0854	−0.0380
Many cancelled consultations/treatments/contacts from relatives	−0.052	0.0120	−4.34	0	−0.0755	−0.0286
Less care after discharge/treatment/stay/contact	−0.045	0.0119	−3.78	0	−0.0686	−0.0218
Has lost important support services	−0.053	0.0120	−4.38	0	−0.0761	−0.0291
Has lost auxiliary services	−0.047	0.0120	−3.96	0	−0.0709	−0.0239
Does not show up for consultations due to fear of becoming infected	−0.054	0.0120	−4.52	0	−0.0780	−0.0308
Stopped taking medications because they are immunosuppressive	−0.003	0.0126	−0.27	0.787	−0.0282	0.0214
Several acute situations have emerged	−0.048	0.0121	−3.97	0	−0.0718	−0.0244
Problems with access to health commodities for chronic illness	−0.022	0.0121	−1.83	0.068	−0.0456	0.0016
Problems with access to medication	0.009	0.0120	0.73	0.466	−0.0148	0.0323

## Data Availability

The data presented in this study are available on request from the corresponding author. The data are not publicly stored.

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
