# Peer review of "Consequences of the Early Phase of the COVID-19 Pandemic for Home-Healthcare Recipients in Norway: A Nursing Perspective"

_healthcare, 2023, doi:10.3390/healthcare11030346_

Round 1
Reviewer 1 Report
This manuscript presents very interesting and important result for understanding healthcare services to COVID-19 patients in their home that needs to be widely circulated. I think this paper is suitable for publication in Healthcare.
I think it would be easier to understand if there was more detailed information about the prognosis, aftereffects, duration of treatment, and other objective results compared to the Likert scale results. Also, what was the most necessary treatment for patients with OVID19 infection obtained from the results of this analysis?
Author Response
Thank you for your comments. We agree that it would be interesting to know more about the themes you call for. Unfortunately, we do not have information about prognosis, treatment etc. of patients with COVID-19 infection in this study. Our study aimed to say something about consequences for home care recipients at one particular point of time, in view of the nurses working in the services. So given, the study design, we are unfortunately not able to add knowledge about what you suggest.
Reviewer 2 Report
The article raises important issues of missed care related to the covid period, which is currently the subject of many studies in nursing.The topic raised in the publication could also be developed in the context of undelivered care/ missed care.
Author Response
Thank you for this comment. We agree that the 'missed care perspective' is relevant for this article. Missed care is a consequence of restrictions during the pandemic for many of the home care service recipients. We have added text about missed care in the background (lines 133-138) and in the discussion (lines 361-368).
Reviewer 3 Report
The article is interesting and proposes very interesting results on the topic of home care nursing during the pandemic. I think it has potential also behind the context of covid. I have some suggestions. I list them in the following comments.
Introduction: It reads very well. I have only some suggestions:
I think to map the COVID situation and to consider the very different studies made on it you could also consider to add the scoping review by Toscano, Tommasi & Giusino (2022).
Line 58: I wonder whether the importance of home healthcare could be extended to general times and not only in crisis. I think that the aging of the population, quality of home care vs hospital care, etc could be good reasons for the healthcare system to improve home care.
Method: I have some questions for you in the method section
How did you provide inputs for questions in the first phase of the questionnaire development? Which procedure has been followed precisely?
How many items have you developed at the begging? Why have some been discarded? Can you be more precise with it?
Can you add indices of the reliability of the developed measure?
REsults: I am okay with that.
Discussion: I am okay with that too.
ONE LAST QUESTION: can you add the developed questionnaire in the Appendix? This could be very helpful for future research and for researchers interested in this topic.
Author Response
Please, see the attachment

Reviewer 4 Report
Minor revision needed

Author Response
Please, see the attachment
